# SHF: Symmetrical Hierarchical Forest with Pretrained Vision Transformer Encoder for High-Resolution Medical Segmentation

**Enzhi Zhang**[1,2]    **Peng Chen**[2] *    **Rui Zhong**[1]    **Du Wu**[2]    **Isaac Lyngaas**[3]
**Jun Igarashi**[2]    **Xiao Wang**[3]    **Masaharu Munetomo**[1]    **Mohamed Wahib**[2]

[1]Hokkaido University, Sapporo, Japan
[2]RIKEN Center for Computational Science, Hyogo, Japan
[3]Oak Ridge National Laboratory, Tennessee, USA
{zhangenzhi, zhongrui, munetomo}@iic.hokudai.ac.jp
{peng.chen, du.wu, jigarashi, mohamed.attia}@riken.jp
{lyngaasir, wangx2}@ornl.gov

## Abstract

This paper presents a novel approach to addressing the long-sequence problem in high-resolution medical images for Vision Transformers (ViTs). Using smaller patches as tokens can enhance ViT performance, but quadratically increases computation and memory requirements. Therefore, the common practice for applying ViTs to high-resolution images is either to: (a) employ complex sub-quadratic attention schemes or (b) use large to medium-sized patches and rely on additional mechanisms within the model to capture the spatial hierarchy of details. We propose Symmetrical Hierarchical Forest (SHF), a lightweight approach that adaptively patches the input image to increase token information density and encode hierarchical spatial structures into the input embedding. We then apply a reverse depatching scheme to the output embeddings of the transformer encoder, eliminating the need for convolution-based decoders. Unlike previous methods that modify attention mechanisms or use a complex hierarchy of interacting models, SHF can be retrofitted to any ViT model to allow it to learn the hierarchical structure of details in high-resolution images without requiring architectural changes. Experimental results demonstrate significant gains in computational efficiency and performance: on the PAIP WSI dataset, we achieved a $3\sim32\times$ speedup or a $2.95\%\sim7.03\%$ increase in accuracy (measured by Dice score) at a $64K^2$ resolution with the same computational budget, compared to state-of-the-art production models. On the 3D medical datasets BTCV and KiTS, training was $6\times$ faster, with accuracy gains of $6.93\%$ and $5.9\%$, respectively, compared to models without SHF.

## 1 Introduction

Recently, Transformers have been rapidly adopted in the field of computer vision [1, 2]. Building on the self-attention mechanism, Vision Transformers (ViTs) and their variants have achieved significant advancements in various image classification and downstream visual tasks [3–7]. Text tokens are atomic, semantically distinct, and rich in information, whereas visual tokens are geometrically related and sparse in semantics. In other words, feeding a sequence of image patches to a transformer encoder deprives the self-attention mechanism of direct information about spatial hierarchy.

---

*Corresponding author.

39th Conference on Neural Information Processing Systems (NeurIPS 2025).

The loss of spatial hierarchy information becomes more pronounced when working with high-resolution or multi-dimensional medical images, as the spatial hierarchy becomes more detailed and intricate [8]. This requires the use of very small patches so that the self-attention mechanism can capture local features [9]. However, using smaller patches quadratically increases the computational and memory costs of self-attention, prompting several approaches to address this issue by modifying the model architecture to help it learn the hierarchical structure in images. Most of these approaches fall into two broad categories: *model-hierarchical* and *attention-hierarchical*. Model-hierarchical solutions involve training ViTs hierarchically, with multiple transformers operating at different resolution levels [10, 8, 11, 12]. While this approach can improve model performance, it also increases training time and memory usage. Additionally, managing multiple interacting transformers adds complexity, requiring extensive hyperparameter tuning at each resolution level. Attention-hierarchical solutions alter the patching scheme at the self-attention stage to represent hierarchical features, as seen in Swin [13, 14], MViT [15], and MViTv2 [16]. Although these methods are more parameter-efficient than standard ViTs [4], they introduce additional spatial operations, increasing model complexity and reducing multi-modal capabilities.

From the above summary, two questions arise: **(1)** How can we design a patching strategy that uses the fewest possible patches to represent the original image, thus increasing the information density of each patch to maintain model performance, without altering the structure of the ViT model? **(2)** If an effective patching strategy could be developed to relay spatial hierarchical information to the model, could we leverage a post-encoder adaptive dispatching strategy during inference to eliminate the need for post-encoder convolution decoding?

To address the two questions above, this paper proposes Symmetrical Hierarchical Forest (SHF), which adapts the patching strategy to the hierarchical details of each training example. This approach enables the ViT to capture hierarchical local spatial information typically derived either from post-encoder convolutions (e.g. the convolution decoder used in the SAM 2 model [17]) or hierarchical architecture features (i.e. Swin [14] and HIPT [8]). We downscale patches in regions with fewer details, aligning them to the patch size used for regions with more details. However, without expert knowledge of hyperparameters, we rely solely on hierarchical forests to extract/represent the spatial hierarchy information. Using the SHF scheme, we demonstrate that the model receives sufficient information about the spatial hierarchy, allowing us to eliminate additional model components (such as U-Net [18] or convolution decoding blocks) that would otherwise be required. Notably, the use of additional components (e.g. the convolution decoder in SAM 2 [17]) can lead to high memory demands to store activations for high-resolution masks (e.g. over 20GB of memory for storing mask activations when using SAM 2 on $64K^2$ images). By employing a transformer encoder-only design with hierarchical forest and post-depatching, we can use smaller patch sizes, enabling self-attention to capture the spatial hierarchy more effectively than larger patches and additional mechanisms (such as U-Net or convolution decoding blocks). As an added benefit, our method simplifies model design and allows for swapping in different encoders, as it is a data-based approach that operates on the input and output of the transformer encoder without modifying the encoder itself.

The contributions of this work are as follows:

- **Symmetrical Hierarchical Forest**. By applying the Symmetrical Hierarchical Forest (SHF), we can extract the hierarchical information directly and eliminate the expert knowledge required for tuning hyperparameters. Additionally, we completely discard the convolution decoder, significantly reducing the computational and memory overhead ($\sim$75% GPUs) of mask processing in high-resolution segmentation tasks. Finally, by downscaling redundant regions in the input image space, we achieve a quadratic reduction in the computational cost of the ViT encoder.

- **Long Context Segmentation**. To demonstrate the efficiency of SHF, we conducted experiments on high-resolution medical imaging datasets, retrofitting state-of-the-art (SoTA) models such as SAM 1 [19] and SAM 2 [17] with our SHF scheme in place of their convolution decoders. When comparing models retrofitted with SHF to SoTA models without SHF, high-resolution pathology datasets (e.g. the PAIP dataset, ranging from $512^2$ to $64K^2$ pixels) and 3D MRI datasets (BTCV, KiTS) benefit from the efficiency of SHF, allowing patch sizes as small as $2\times2$ pixels and $2\times2\times2$ voxels. At the same performance level, we achieve a $3\times$ to $32\times$ speedup on the PAIP dataset, or, with the same computational budget, a 7.03% increase in Dice score at $64K^2$ resolution. On 3D medical imaging datasets, such as BTCV and KiTS, we see a $\sim$6$\times$ training speed improvement along with performance increases of 6.93% and 5.9%, respectively, compared to SoTA models without SHF.

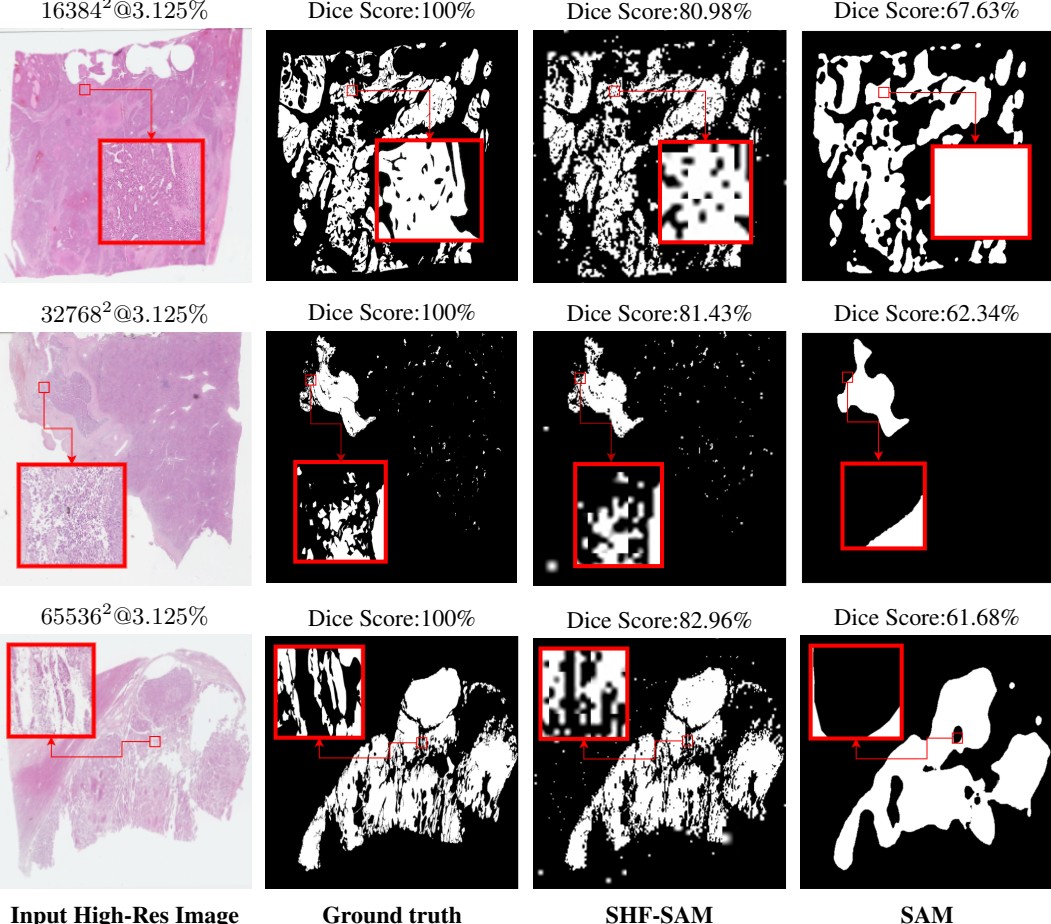

| | | | |
|---|---|---|---|
| $16384^2@3.125\%$ | Dice Score:100% | Dice Score:80.98% | Dice Score:67.63% |
| $32768^2@3.125\%$ | Dice Score:100% | Dice Score:81.43% | Dice Score:62.34% |
| $65536^2@3.125\%$ | Dice Score:100% | Dice Score:82.96% | Dice Score:61.68% |
| **Input High-Res Image** | **Ground truth** | **SHF-SAM** | **SAM** |

Figure 1: We compared the segmentation difference between SAM [19] and SAM retrofitted with our SHF scheme (SHF-SAM) instead of the original convolution decoder for PAIP [20] at $16,384^2, 32,768^2$, and $65,536^2$ resolutions. At the same GPU budgets, SHF-SAM can go down to batch size of $8x8$ (vs. $1,024x1,024$ at best for SAM before going OOM). As a result, SHF-SAM can extract and express mask details better than SAM, with the gap in accuracy favoring SHF-SAM as the resolution gets higher.

- **Simplicity and Low-Overhead**. Unlike existing methods that modify the self-attention or transformer encoder mechanisms, our solution preserves the original self-attention mechanism. This ensures seamless retrofitting of SHF into any vision transformer. SHF is a low-overhead pre-processing and post-processing solution that is further amortized over epochs, making the overhead effectively negligible.

The rest of this paper is organized as follows: Section 2 reviews related work. Section 3 presents the methodology. Section 4 describes the experimental setup. Section 5 reports the evaluation results. Finally, Section 6 concludes the paper and outlines future work.

## 2 Related Work

The computational cost of self-attention increases as the patch size decreases. To mitigate this, several strategies have been developed. Sequence parallel methods, including Deep-Speed Ulysses [21], LightSeq [22], RingAttention [23], LLS [24], FlashAttention [25], and [26]. Linear approximation methods, such as spectral attention[27–29], low-rank approximation [30, 31], sparse attention matrix sampling [32–36], infrequent self-attention updates [37, 38], or combinations of these [39]. These methods reduce the computational load of the attention mechanism; however, excessive reduction

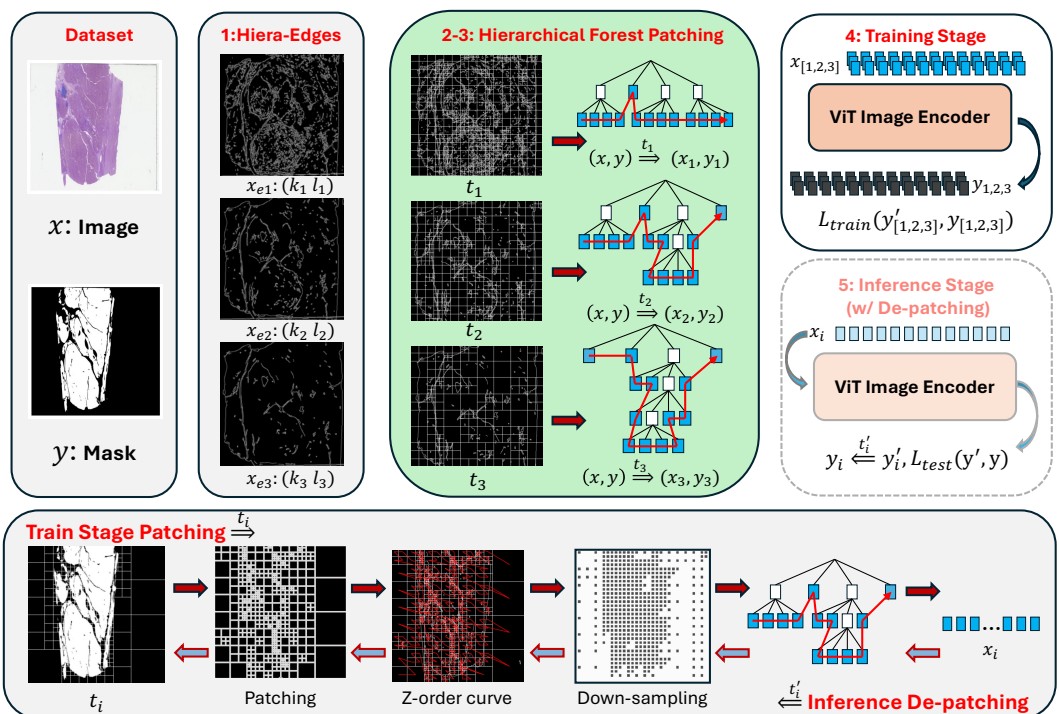

Figure 2: Overview of SHF. SHF begins with the original image and ends with feeding the extracted patches (tokens) into an intact transformer-based model. In a real training example on the SAM [19] model using 512×512 images from the PAIP [20] liver cancer dataset, SHF reduces the number of patches from 4,096 to 512 (each of size 4×4) while maintaining the same Dice score. This results in an ∼8× reduction in sequence length and a ∼7.53× speedup in end-to-end training.

can lead to performance loss, as reported in the literature [40]. Hierarchical training of ViTs, where multiple transformers are trained at different resolution levels [10, 8, 11, 12]. However, using multiple transformers increases training time and memory usage, and managing multiple interacting transformers is complex. Recently, quadtrees have been used in image segmentation to reduce attention cost, e.g. quadtree/octree attention or patch pre-processing [41–43]. Both of those approaches employ quadtrees, but involve additional model complexity or need expert knowledge in the processing stage for hyperparameters.

## 3 Methodology

### 3.1 Vision Transformers and Attention

The self-attention mechanism in transformers computes attention scores $A$ between input tokens, forming the attention matrix. Let $x \in R^{N \times F}$ denote a sequence of $N$ feature vectors of dimensions $F$. A transformer is a function $\mathcal{T} : R^{N \times F} \to R^{N \times F}$ defined by the composition of $L$ transformer layers $\mathcal{T}_1(\cdot), ..., \mathcal{T}_L(\cdot)$ as follows:

$$Tr_l(x) = f_l(A_l(x) + x) \tag{1}$$

$A_l(\cdot)$ is the self-attention function. The function $f_l(\cdot)$ transforms each feature independently of the others and is usually implemented with a small two-layer feedforward network. Formally, the input sequence $x$ is projected by three matrices $W_Q \in R^{F \times D}, W_K \in R^{F \times D}$, and $W_V \in R^{F \times D}$, to corresponding representations $Q$, $K$ and $V$. Thus, the attention scores are calculated as follows:

$$Q = xW_Q, K = xW_K, V = xW_V, A_{ij} = \text{Softmax}\left((Q_i K_j^T)/\sqrt{d_k}\right) \tag{2}$$

where $Q_i$ and $K_j$ are query and key vectors for tokens $i$ and $j$, and $d_k$ is the dimension of the key vectors. The complexity of the attention matrix is $O(N^2)$, where $N$ is the sequence length. We further assume that the input is the content of a square image $x$ with a resolution of $Z$, that is, let

Table 1: Speedup of SHF end-to-end training for PAIP dataset at the same segmentation quality as the baseline. We use the highest dice score of the baseline models, SAM and SHF-SAM.

| Resolution | Model-Patch | Seconds/Image | GPUs | Sequence Length | Dice Score (%) | Speedup ($\times$) |
|---|---|---|---|---|---|---|
| $512 \times 512$ | **SHF-SAM-16** | 0.03616 | 1 | 256 | 71.03 | 7.75$\times$ |
| | SAM-4 | 0.28041 | 1 | 16,384 | 68.98 | |
| $1,024 \times 1,024$ | **SHF-SAM-16** | 0.09913 | 1 | 1,024 | 73.65 | 3.86$\times$ |
| | SAM-8 | 0.38261 | 8 | 16,384 | 66.56 | |
| $4,096 \times 4,096$ | **SHF-SAM-16** | 0.1031 | 8 | 1,024 | 74.17 | 15.7$\times$ |
| | SAM-32 | 1.6183 | 64 | 16,384 | 71.05 | |
| $8,192 \times 8,192$ | **SHF-SAM-16** | 0.3512 | 16 | 2,048 | 76.21 | 7.17$\times$ |
| | SAM-64 | 2.5168 | 128 | 16,384 | 67.31 | |
| $16,384 \times 16,384$ | **SHF-SAM-16** | 0.3672 | 32 | 2,048 | 76.89 | 15.4$\times$ |
| | SAM-128 | 5.6714 | 256 | 16,384 | 67.63 | |
| $32,768 \times 32,768$ | **SHF-SAM-32** | 0.3826 | 64 | 2,048 | 76.08 | 23.8$\times$ |
| | SAM-256 | 9.1213 | 512 | 16,384 | 62.34 | |
| $65,536 \times 65,536$ | **SHF-SAM-64** | 0.4013 | 256 | 2,048 | 75.33 | 32.3$\times$ |
| | SAM-1024 | 12.9833 | 1,024 | 16,384 | 61.68 | |

$x \in R^{Z \times Z}$, and by assuming that patches arise from the uniform grid patch method of patch size $p$. Thus the sequence $N = (Z/P)^2$. The total computation and memory cost of attention scores defined in Eqn 2 according to resolution and patch size is $O([Z/P]^4)$. This complexity shows the difficulties of increasing the resolution while decreasing the patch size $P$ with the uniform grid patch strategy.

## 3.2 Symmetrical Hierarchical Forest (SHF)

In the following paragraphs, we describe how SHF works, following the steps outlined in Fig. 2:

**Hiera-Edges Detection**. We aim to use different methods to extract hierarchical details, as this would allow us to augment the dataset. This, however, is different from traditional augmentation applied at the image level; we augment by providing the model with different views of the spatial hierarchy for each image, thereby giving the model a better opportunity to learn the hierarchy. To use different ways to extract the hierarchical details in the image $x \in X$, as Fig. 2-1, we use a Gaussian blur with kernel $k$ and Canny [44] edge detection with threshold $l$ to the original input images $x \in X$. The Gaussian blur with kernel $k$ smooths the irrelevant details, and the Canny edge detection with threshold $l$ extracts the grayscale edges $x_e$ of the image. To generate different spatial structures that can also recover the original inputs, during our experiments, we randomly choose the threshold to be in the range [100,200], and the kernel size is randomly set to one of [3,5,7,9,11,13].

**Hierarchical Forest**. In the Hierarchical Forest stage (Fig. 2-2), we build several quadtrees (octrees in 3D) from each $x_e$; $x_e$ undergoes a recursive tree partitioning. To construct the tree $T$, we create tree nodes $T_n$ representing specific regions where $n$ is the number of leaf nodes. The leaf nodes $T_{n+s-1}$ is defined recursively as follows:

$$T_{n+s-1} = \begin{cases} T_n, & \text{if } n \geq N \\ T_n[i] = \{T_n^1, T_n^2, ..., T_n^s\}, & \arg\max_i V(T_n[i]) \end{cases} \quad (3)$$

where $V$ is the criterion we use to differentiate the different levels of detail. We choose the maximum sum of pixel values among the tree nodes by $i = \arg\max_i V(T^n[i])$, $\{T_n^1, T_n^2, ..., T_n^s\}$ are the $s$ new child nodes after the subdivision of $T^n[i]$, $s = 2^d$ is the number of subdivisions and $d$ is the number of dimensions. $T$ can be used for both 2D and 3D tasks, for example $s = 4, 8$ which means quadtree and octree, respectively [45, 46]. In our implementation, to avoid unnecessary padding or dropping due to varying sample lengths, we control the number of splits at the leaf nodes to keep each sample at the same sequence length $N$. This approach fully utilizes the GPU resources without discarding sample information. The sequence length $N$ is set to $[1024, 4096, 8192, 16384, 16384, 16384]$ with respect to resolutions, which practically allows the input $x_e$ to be subdivided all the way down to the $2 \times 2$ patch size level.

After building the tree, as a property of quad/octrees, visiting all the tokens (appearing as leaves in the tree) from left to right gives the token sequence as a Z-order space-filling curve in the image space [47]. This operation($\xrightarrow{t_i}$) results in a sequence of image patches shown in step Fig. 2-3. We

not only use spatial trees to encode images $x$, but we also encode masks $y$. Using the same tree and z-order of the mask will also result in a sequence of mask patches $y_i$. Since the spatial trees are built from the image, the sequence length of the image and mask patches $(x_i, y_i)$ should be the same $N$.

**Image Encoder**. After completing hierarchical forest patching, we obtain an image patch sequence and a mask patch sequence of the same length, allowing us to (continuously) train the ViT model. We use the well-known pre-trained image encoder from SAM in steps Fig. 2-4 and then continue training the model on our dataset after retrofitting our SHF scheme into the SAM model. SAM uses an MAE [2] pre-trained ViT [4], minimally adapted to process high-resolution inputs. For 3D MRI data tasks, we use the SAM 2 [17] encoder and similarly retrofit SHF into SAM 2. When retrofitting SHF into both SAM and SAM 2, we remove the convolution decoder that is part of the original SAM and SAM 2 models. It should be noted that during the training phase of the model, we do not directly generate masks $y'$ but instead generate encoded masks $y'_i$. The advantage of this approach is that it saves memory, which would otherwise be needed for high-resolution masks, and reduces the backpropagation compute costs associated with the heavy decoders. We summarized the objective function as follows:

$$\min_{\theta} \mathbb{E}_{\substack{(x,y)\sim\mathcal{D} \\ k\sim\mathcal{K}, l\sim\mathcal{L}}} \left[ \sum_{i=1}^{M} \mathcal{L}_{\text{hiera}}\Big( f_\theta\big(T_{\text{tree}}(x, k, l)_i\big),\ T_{\text{sym}}(y, k, l)_i \Big) \right] \tag{4}$$

Where $M$ is total number of samples, $\theta$ represents the trainable weights, $\mathcal{L}_{\text{hiera}}$ denotes the Dice loss, $(k, l)$ specify the Gaussian filter size and Canny threshold, and $T_{tree} = T_{sym}$ are the recursive tree operators applied to both the input $x$ and the mask $y$.

**Symmetrical Depatching**. The function of depatching ($\overset{t'_i}{\Longleftarrow}$) is to upscale the sequence obtained from the ViT into a mask during the inference stage. By depatching, we can eliminate the need for general U-Net or lightweight decoders and calculate the backpropagation gradient directly at the output of the encoder (in Fig. 2-5). This is because we have observed that the information in the mask itself is sparser compared to the input image (see the Fig. 4). Therefore, when reconstructing the mask at the evaluation stage, we simply use the same quad/octree structure derived from the image to linearly upscale the patches at the corresponding positions in the sequence.

## 4 Experimental Setup

### 4.1 High-Resolution Medical Image Datasets: PAIP, BTCV, & KiTS

**PAIP:** [20] is a high-resolution, real-world liver cancer pathology dataset, with sample resolutions up to $64K^2$, significantly surpassing those of conventional image datasets. PAIP contains $2,457$ Whole-Slide Images (WSIs). When lower resolutions are needed, we downscale the images to uniform sizes of $[512, 1024, 4096, 8192, 16384, 32768, 65536]$ square pixels. For training, we randomly select $70\%$ of samples, $10\%$ for validation, and $20\%$ for testing. All datasets are shuffled and normalized to $[0.0, 1.0]$ as input for the model. The **BTCV:** challenge [48] for 3D multi-organ segmentation includes 30 subjects with abdominal Computed Tomography (CT) [49] scans, with 13 organs annotated by experts. Each CT scan consists of 80 to 225 slices, each with $512^2$ pixels. The multi-organ segmentation task is defined as a 13-class segmentation problem, where the average dice score across all classes is typically reported. Although BTCV has a lower resolution compared to the PAIP dataset ($512^2$ vs. $64K^2$), it remains a widely used benchmark in the high-resolution medical segmentation community. **KiTS:** the 2019 Kidney and Kidney Tumor Segmentation Challenge (KiTS19 [50]) aimed to develop algorithms for segmenting kidneys and kidney tumors in contrast-enhanced 3D CT scans. The dataset consists of 300 anonymized scans in NIfTI (.nii.gz) format, each manually annotated with kidney and tumor labels. Each scan has a typical resolution of $512\times512$ pixels per slice, with the number of slices varying based on patient anatomy. KiTS is a common benchmark for 3D MRI segmentation [51].

### 4.2 Evaluating Models: Baselines & Proposed

**Baseline Model:** We use the well-known segmentation model SAM [19], which employs a Vision Transformer (ViT) encoder as its backbone for segmentation tasks. SAM offers ViT variants (ViT-Base (b), ViT-Large (l), and ViT-Huge (h)), each with 12, 24, or 32 transformer layers, respectively.

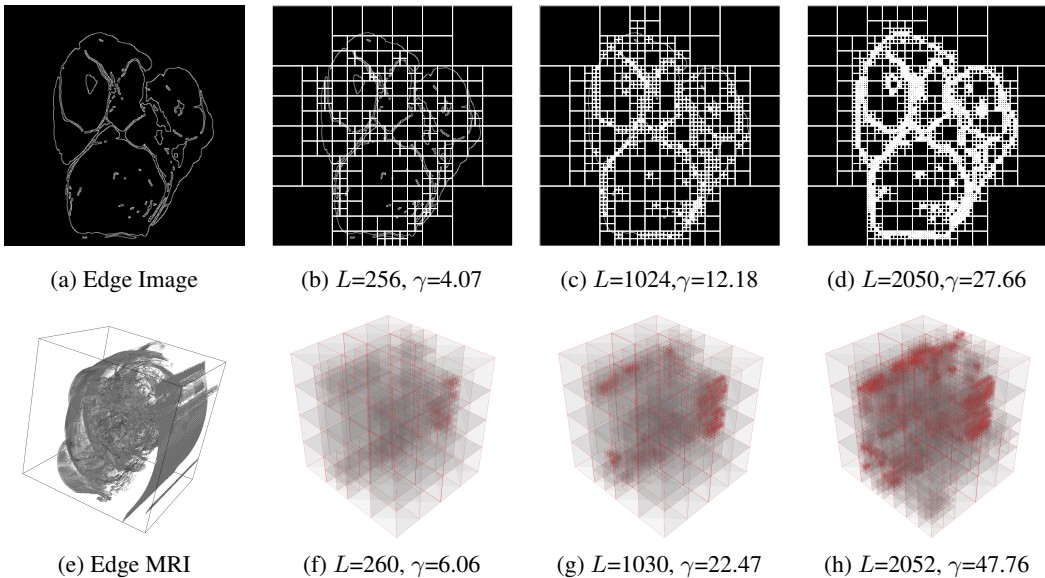

(a) Edge Image     (b) $L$=256, $\gamma$=4.07     (c) $L$=1024,$\gamma$=12.18     (d) $L$=2050,$\gamma$=27.66

(e) Edge MRI     (f) $L$=260, $\gamma$=6.06     (g) $L$=1030, $\gamma$=22.47     (h) $L$=2052, $\gamma$=47.76

Figure 3: Spatial subdivision of an edge image with a resolution of $1024\times1024$. $L$ represents the sequence length, and $\gamma$ denotes the compression ratio relative to the grid patches. SHF significantly compresses the image, with higher dimensions resulting in a higher compression ratio.

These weight configurations -b, l, and h- are pre-trained using Masked Autoencoders (MAE) [2]. Unlike Unet-style models [52, 18, 53, 13], the SAM mask decoder is lightweight, containing only two convolutional layers. However, reconstructing high-resolution masks from the latent space requires additional upscaling layers, leading to increased memory usage.

**Proposed Model:** As defined in Eqn 4, our approach utilizes the tree structure from the patching stage in SHF, reducing the need for decoder training. We also experimented with the updated SAM2 [17] encoder ViT model, adapting it for 3D MRI data tasks, such as BTCV and KiTS19. Unlike SAM2 original video frame processing, we use 3D convolutional layers for voxel processing in the patch embedding stage. Unet-shaped models, such as UNETR [18], TransUnet [53], and SWIN Unet [13], follow a contraction-expansion pattern with transformer or convolutional layers as encoders to extract image details, connected to decoders via skip connections. These decoders upscale the representation vectors from the dense latent space to match the mask's size. For comparison with SHF, we test Unet [52] with pre-trained weights from the timm package, along with the SAM-pretrained UNETR [18] and TransUnet [53] models.

**Performance Metrics:** Computational performance is reported in seconds per image for end-to-end training. Segmentation accuracy is evaluated using the Dice score, which quantifies the overlap between predicted and ground truth segmentation masks. It is defined as:

$$\text{Dice(X,Y)} = 2 \cdot |X \cap Y|/(|X| + |Y|) \tag{5}$$

where $X$ and $Y$ are the sets being compared, and $|X \cap Y|$ is the size of their intersection.

## 5 Evaluation

### 5.1 Speedup: SHF vs w/o SHF on the same models, datasets, and comparable performance

Training with SHF is faster due to its ability to significantly compress the sequence length , shown in Figure. 3, and eliminate the decoder component. As shown in Table 1, SHF, which combines pre-processing and post-processing steps on top of the pre-trained SAM baseline, achieves a geometric mean speedup ranging from $3.86\times$ to $32.3\times$ while maintaining comparable dice scores. This speedup is measured when both SHF and the baseline are trained for the same number of epochs. At the highest resolution of $64K^2$ with training on 256/1024 GPUs, SHF delivers approximately a $32\times$ speedup compared to naive SAM encode and a reduction of $75\%$ usage of GPUs. Such a speedup

Table 2: Improvement in quality of segmentation (IQS) for the PAIP dataset against different baselines.

| Resolution | Model | Patch | Seconds/Image/GPU | GPUs | Sequence Length | Dice Score (%) |
|---|---|---|---|---|---|---|
| $1,024 \times 1,024$ (IQS: **+2.95%**) | SAM | $8^2$ | 0.3826 | 8 | 16,384 | 66.56 |
| | UNETR | $8^2$ | 1.0863 | 32 | 16,384 | 75.72 |
| | TransUNet | - | 1.3247 | 8 | - | 72.38 |
| | UNet | - | 0.0981 | 1 | - | 68.92 |
| | **SHF+SAM** | $2^2$ | 0.0991 | 1 | 1,024 | **78.67** |
| $4,096 \times 4,096$ (IQS: **+3.22%**) | SAM | $32^2$ | 1.6183 | 64 | 16,384 | 71.05 |
| | UNETR | $32^2$ | 1.8613 | 128 | 16,384 | 75.77 |
| | TransUNet | - | 2.1637 | 128 | - | 71.32 |
| | UNet | - | 0.3712 | 16 | - | 64.11 |
| | **SHF+SAM** | $2^2$ | 0.3766 | 8 | 4,096 | **78.99** |
| $8,192 \times 8,192$ (IQS: **+4.41%**) | SAM | $64^2$ | 2.5168 | 128 | 16,384 | 67.31 |
| | UNETR | $64^2$ | 2.6618 | 256 | 16,384 | 75.27 |
| | TransUNet | - | 2.3678 | 256 | - | 70.89 |
| | UNet | - | 1.2858 | 32 | - | 63.21 |
| | **SHF+SAM** | $2^2$ | 1.5327 | 16 | 8,192 | **79.68** |
| $16,384 \times 16,384$ (IQS: **+5.09%**) | SAM | $128^2$ | 5.6714 | 256 | 16,384 | 67.63 |
| | UNETR | $128^2$ | 5.1179 | 512 | 16,384 | 75.89 |
| | TransUNet | - | 6.1296 | 512 | - | 70.46 |
| | UNet | - | 2.7825 | 256 | - | 62.97 |
| | **SHF+SAM** | $2^2$ | 3.2741 | 32 | 16,384 | **80.98** |
| $32,768 \times 32,768$ (IQS: **+6.47%**) | SAM | $256^2$ | 9.1213 | 512 | 16,384 | 62.34 |
| | UNETR | $256^2$ | 8.1896 | 1,024 | 16,384 | 74.96 |
| | TransUNet | - | 10.001 | 1,024 | - | 69.88 |
| | UNet | - | 4.2714 | 512 | - | 61.38 |
| | **SHF+SAM** | $4^2$ | 3.4631 | 64 | 16,384 | **81.43** |
| $65,536 \times 65,536$ (IQS: **+7.03%**) | SAM | $1,024^2$ | 12.983 | 1,024 | 16,384 | 61.68 |
| | UNETR | $1,024^2$ | 13.218 | 2,048 | 16,384 | 75.31 |
| | TransUNet | - | 14.352 | 2,048 | - | 67.67 |
| | UNet | - | 5.9611 | 1,024 | - | 59.69 |
| | **SHF+SAM** | $8^2$ | 3.6112 | 256 | 16,384 | **82.96** |

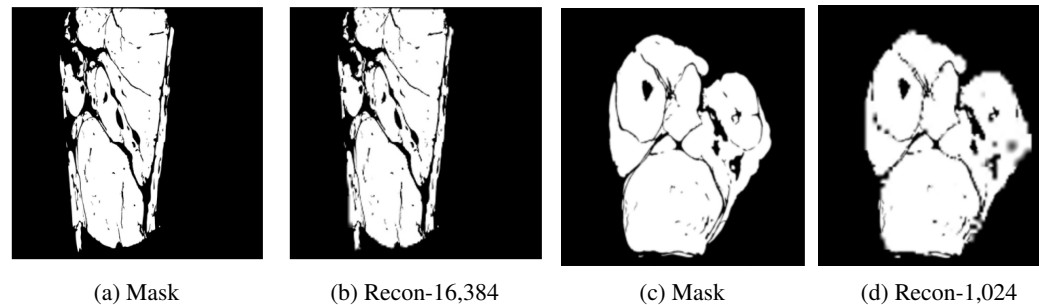

(a) Mask      (b) Recon-16,384      (c) Mask      (d) Recon-1,024

Figure 4: Reconstructed mask through the SHF build from the input $64K^2$ image with sequence length $16,384$ and $1,024$ with recovery accuracy $99.5\%$ and $93.7\%$. These high recovery accuracies suggest the information contained in the masks are sparse.

benefited from eliminating the heavy decoder, allowing more parallelization and reducing inference and back-propagation.

## 5.2 Segmentation Performance: SHF vs w/o SHF on the different models and datasets

**Qualitative Results:** Table 2 demonstrates the segmentation improvements across different models and PAIP resolutions. At comparable resolutions, SHF achieves a nearly $8\times$ reduction in patch size while maintaining the same computational complexity. This results in an average $5.5\%$ improvement in dice score over the original model. Additionally, these improvements come with training time speedups of up to $4.6\times$. At high resolution ($64K^2$), SHF lacks a decoder during training, offering substantial computational and GPU memory savings compared to SAM, which relies on an upsampling restoration mask. Consequently, SAM requires a shorter sequence length at high resolutions, as it performs more upsampling to generate masks. For instance, as shown in Table 1, moving from $16K^2$ to $64K^2$ resolution results in a slight $2\%$ performance drop for SAM. Table 3 presents 3D MRI

Table 3: Segmentation of BTCV [48] for multi-organ segmentation and KiTS19 [50] for Kidney Tumor Segmentation on a single GPU. *Time* indicates the end-to-end runtime to achieve the corresponding Dice Score.

| Datset | Model | Patch Size | Time | Speedup ($\times$) | Dice Score (%) |
|---|---|---|---|---|---|
| BTCV [48] | U-Net [52] | N/A | 843.90 Seconds | 9.04$\times$ | 80.2 |
| | U-Mamba [52, 54] | N/A | 8,016.24 Seconds | 0.95$\times$ | 83.51 |
| | TransUNet [55] | N/A | 3115.25 Seconds | 2.45$\times$ | 83.8 |
| | UNETR [18] | $4^3$ | 8386.56 Seconds | 0.91$\times$ | 89.1 |
| | Swin UNETR [56] | $4^3$ | 5861.93 Seconds | 1.30$\times$ | 89.5 |
| | SAM2[17] | $4^3$ | 7637.28 Seconds | 1.0$\times$ | 82.77 |
| | **SHF-SAM2** | $2^3$ | **1067.88 Seconds** | **7.15$\times$** | **89.71** |
| KiTS [50] | U-Net [52] | N/A | 243.7 Minutes | 1.99$\times$ | 83.23 |
| | U-Mamba [52, 54] | N/A | 969.0 Minutes | 0.5$\times$ | 86.22 |
| | CoTr [55] | N/A | 488.7 Minutes | 0.99$\times$ | 84.59 |
| | UNETR [18] | $8^3$ | 513.6 Minutes | 0.94$\times$ | 86.45 |
| | nnFormer [51] | $8^3$ | 876.5 Minutes | 0.55$\times$ | 75.85 |
| | Swin UNETR [51] | $8^3$ | 748.3 Minutes | 0.65$\times$ | 81.27 |
| | Swin UNETR-V2 [51] | $8^3$ | 766.4 Minutes | 0.63$\times$ | 84.14 |
| | SAM2[17] | $8^3$ | 483.1 Minutes | 1.0$\times$ | 81.35 |
| | **SHF-SAM2** | $2^3$ | **187.3 Minutes** | **2.58$\times$** | **87.25** |

segmentation results for BTCV and KiTS datasets at a $512^3$ resolution. Instead of processing each 2D slice independently and reconstructing the final 3D prediction as in previous works [57, 53], we replace the 2D convolution patch embedding with 3D convolution for voxel processing. As shown in the tables, SHF outperforms the prestrained SAM2 Hiera image encoder, yielding a 5.9% higher dice score and requiring 4$\times$ less computational resources.

**Visual Results:** We compare the segmentation quality across different high resolutions ($[16K^2, 32K^2, 64K^2]$) for the baseline models SAM, and our proposed SHF. The segmentation results are summarized in Fig. 1. The first column shows the original input, with the label indicating the resolution. The red square highlights a small portion of the image, approximately 3.125% of the entire slide. The second column presents the ground truth, followed by the predictions from different models. At higher resolutions, all models can capture the general segmentation areas. However, due to the limitations of heavy convolution-based decoders and uniform grid patching, only large patch sizes are feasible, such as the $16K^2$ patch size used by SAM and UNETR at a $64K^2$ input resolution. In contrast, at the same $64K^2$ resolution, SHF can use smaller patch sizes, as small as $8^2$, significantly improving the quality of the detailed masks.

### 5.3 SHF vs HIPT hierarchical model: training from scratch on the same datasets

To demonstrate the versatility of SHF, we compare its classification performance on the PAIP dataset with HIPT [8], a SoTA highly advanced hierarchical multi-resolution model specifically designed for microscopic pathology classification. For this experiment, we restructured the PAIP dataset, originally intended for segmentation, into six organ-based categories. Each category consists of 40 samples, with 28 for training, 8 for testing, and 4 for validation. For HIPT, we resized all samples to three resolution

Table 4: Classification (Top-1 accuracy) of vanilla ViT, HIPT [8], and SHF-ViT on PAIP dataset ($16,384^2$ res.)

| Model | GPUs | Patch Size | Accuracy |
|---|---|---|---|
| ViT [4] | 128 | $4,096^2$ | 68.97 |
| HIPT [8] | 128 | $[16, 256^2, 4,096^2]$ | 72.69 |
| SHF-ViT-4096 | 8 | $4,096^2$ | 69.11 |
| SHF-ViT-2 | 128 | $2^2$ | **80.14** |

scales ($[256, 1024, 16384]$) and set the patch sizes for each scale to $[16, 256, 4096]$, adhering to the original settings. For SHF, we used only the $16K^2$ resolution images for classification. Instead of relying on a decoder for segmentation, we added an output channel dedicated to class prediction. In Table 4, SHF achieves significant accuracy improvement (>8%) over HIPT, even when using a vanilla ViT model and the same computational budget. At high resolution ($16K^2$), HIPT is limited to a patch size of $4096^2$ before running OOM, whereas SHF can use patch sizes as small as $2^2$ in the highest-resolution regions. This substantial accuracy boost, despite SHF using a basic ViT, indicates

that: a) SHF avoids random patch dropping or padding, and b) smaller patch sizes play a more crucial role in improving performance than model complexity.

# 6 Conclusion and Future Work

This paper presents SHF, a lightweight and efficient method for enhancing ViT performance on high-resolution images. By increasing token information density and encoding hierarchical spatial structures through a hierarchical patching strategy and reverse depatching, SHF enables standard ViTs to handle long sequences effectively without requiring architectural changes. Experimental results on both 2D and 3D medical imaging tasks demonstrate significant improvements in computational efficiency and segmentation accuracy, establishing SHF as a practical solution for high-resolution medical image analysis. In future work, we aim to extend SHF to broader scientific domains, including materials discovery [58, 59], brain neuron structure detection [60, 61], and non-medical CT applications [62–64]. Additionally, we plan to enhance SHF's performance on multimodal problems [65] to support more complex and diverse applications.

# 7 Acknowledgment

This manuscript has been co-authored by UT-Battelle, LLC, under contract DE-AC05-00OR22725 with the US Department of Energy (DOE). The U.S. Government retains a nonexclusive, worldwide license to publish or reproduce the published form of this manuscript, or to authorize others to do so, for U.S. Government purposes, as acknowledged by the publisher. This research was partially supported by the ORNL AI Initiative sponsored by the ORNL Director's Research and Development Program. This work was supported by JSPS KAKENHI under Grant Number JP21K17750.

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

# Appendix

## Appendix Contents

# A  Summary of Methods for Training on Long Sequences

Table 5 provides a summary of recent approaches that address the long-sequence challenges encountered by ViT models when processing high-resolution images.

Table 5: A summary of relevant long sequence training methods for solving the quadratic attention through the reduce the amount of work. Here $N$ = sequence length.

| Approach | Method | Merits & Demerits | Complexity (Best) | Model | Implementation |
|---|---|---|---|---|---|
| Attention Approximation | Longformer [35] ETC [66] | (+) Better time complexity vs Transformer.
(-) Sparsity levels insufficient for gains to materialize. | $O(N)$
$O(N\sqrt{N})$ | Some Models w/ Forked PyTorch | Self-attention Implementation |
| | BigBird [67] Reformer [68] | (+) Theoretically proven time complexity.
(-) High-order derivatives | $O(N \log N)$ | Some Models w/ Forked PyTorch | Self-attention Implementation |
| | Sparse Attention [69] | (+) Introduced sparse factorizations of the attention.
(-) Higher time complexity. | $O(N\sqrt{N})$ | Some Models w/ Forked PyTorch | Self-attention Implementation |
| | Linformer [70] Performer [30] | (+) Fast adaptation
(-) Assumption that self-attention is low rank. | $O(N)$ | Some Models w/ Forked PyTorch | Self-attention Implementation |
| | SPFormer [71] (Prediction) | (+) Irregular tokens.
(-) No adaptation to high resolution. | $O(P^2)$
P:num of regions | Model w/ Plain PyTorch | Model Implementation |
| Hierarchical | Hier. Transformer [10] (Text Classification) | (+) Independent hyperpara. tuning of hierarc. models.
(-) No support for ViT. | $O(N \log N)$ | Model w/ Plain PyTorch | Model Implementation |
| | CrossViT [11] (Classification) | (+) Better time complexity vs standard ViT.
(-) Complex token fusion scheme in dual-branch ViTs. | $O(N)$ | Model w/ Plain PyTorch | Model Implementation |
| | HIPT [8] (Classification) | (+) Model inductive biases of features in the hierarchy.
(-) High cost for training multiple models. | $O(N \log N)$ | Model w/ Plain PyTorch | Model Implementation |
| | MEGABYTE [12] (Prediction) | (+) Support of multi-modality.
(-) High cost for training multiple models. | $O(N^{4/3})$ | Model w/ Plain PyTorch | Model Implementation |
| State Space Model | MEGABYTE [12] (Prediction) | (+) Support of multi-modality.
(-) High cost for training multiple models. | $O(N^{4/3})$ | Model w/ Plain PyTorch | Model Implementation |
| High-resolution | xT [12] (Prediction) | (+) Support of multi-modality.
(-) High cost for training multiple models. | $O(N^{4/3})$ | Model w/ Plain PyTorch | Model Implementation |
| | MEGABYTE [12] (Prediction) | (+) Support of multi-modality.
(-) High cost for training multiple models. | $O(N^{4/3})$ | Model w/ Plain PyTorch | Model Implementation |
| | MEGABYTE [12] (Prediction) | (+) Support of multi-modality.
(-) High cost for training multiple models. | $O(N^{4/3})$ | Model w/ Plain PyTorch | Model Implementation |
| **Ours** | **Symmetrical Hierarchical Forest (Segmentation & Class.)** | **(+) Attention mechanism intact.**
**(+) Largely reduces computation cost; maintains quality.**
**(+) Efficiency depends on level of details in an image.**
**(-) task semantics are independent of edge information.** | $O(\log^2 N)$ | Any Model w/ Plain PyTorch | Image Pre-processing |

# B  Discussion and Ablation Studies

## B.1  Sequence length $L$ and compression ratio $\gamma$

Fig. 3 illustrates how different sequence lengths affect the compression ratio of the original image, using the same input image and edge extraction algorithm. The first column displays an edge image with a resolution of $1024 \times 1024$ pixels. We evaluated sequence lengths $L = [256, 1024, 2050]$, which correspond to average patch sizes of $[31.7, 4.3, 9.17]$ pixels under the grid patch configuration. The resulting compression ratios $\gamma$ are $[4.07, 12.18, 27.66]$. For image segmentation, a sequence length of 1024 appears to provide the best balance for images at 1K resolution. In 3D, the compression ratios increase to $[6.06, 22.47, 47.76]$, as higher dimensionality leads to sparser information within each patch, thereby making experiments in higher visual dimensions feasible.

The key reason SHF can handle small patch sizes at high resolutions is its ability to reduce sequence length through hierarchical forest patching. In Fig. 6, we illustrate how the sequence length can be adjusted by tuning the threshold of the split value without substantially affecting prediction performance. The split value $v$ governs both the total sequence length and the distribution of patch sizes. The first row of Fig. 6 shows that halving the split value $[100, 50, 20]$ leads to a roughly proportional change in patch size distribution, with average patch sizes $[12.77, 19.17, 29.88]$. This demonstrates a linear relationship between the split value and the average patch size. In contrast, for the uniform grid patching strategy, sequence length grows quadratically as $O\big((\frac{Z}{P})^2\big)$. Using

hierarchical forest patching, however, we observe an approximately linear increase in average sequence length as the average patch size decreases. In 3D, the average patch sizes $[6.27, 11.79, 17.86]$ are larger due to increased dimensionality, which results in sparser information per patch and makes experiments in higher visual dimensions feasible.

## B.2 Image Information Loss in Patching and Depatching

In Fig. 4, we present the tree structure generated from the input image using the patch partitioning strategy, showing the effects of compressing and reconstructing the original mask. When decoding the mask through spatial partitioning, the accuracy can theoretically reach an upper limit of 99.5% with perfect predictions. However, as our experiment's sequence mask achieves only 82.97%, this tree-structured reconstruction's impact on decoding performance is relatively minor.

Notably, SHF differs from a convolution-based decoder in terms of image quality degradation. We believe this is due to two types of losses: the first is texture loss, where incorrect model regression generates inaccurate textures, leading to noise artifacts in the SHF image. The second is structural loss, observed as an amplification of noise in patches with varying structures introduced during the Dispatching stage. We attribute both losses to the loss of geometric relationships between patches during compression. Fig. 5 illustrates the degradation of these geometric properties.

## B.3 Training the SHF Algorithm: Hyperparameters and Loss

Since depatching happens in the evaluation stage, our algorithm is also divided into the training stage and the evaluation stage. Let's first look at the training stage, where $L$ represents the length of the hierachical patching sequence, $K$ represents the optional kernel size, $t_l$ and $t_h$ represent the lower and higher threshold of canny edge detection, $f$ is the model, $x$ is the input, $\theta$ is the trainable parameter, $D$ is the dataset, $N$ is the number of batches, and $E$ is the total number of epochs required. Here, we provide a Python pseudocode in Code Listing 1 for reference.

```python
def build_hierachical_forest(self):
    h,w = self.domain.shape
    root = Rect(0,w,0,h)
    self.nodes = [[root, root.contains(self.domain)]]
    while len(self.nodes) < self.fixed_length:
        bbox, value = max(self.nodes, key=lambda x:x[1])
        idx = self.nodes.index([bbox, value])
        if bbox.get_size()[0] == 2:
            break

        x1,x2,y1,y2 = bbox.get_coord()
        lt = Rect(x1, int((x1+x2)/2), int((y1+y2)/2), y2)
        v1 = lt.contains(self.domain)
        rt = Rect(int((x1+x2)/2), x2, int((y1+y2)/2), y2)
        v2 = rt.contains(self.domain)
        lb = Rect(x1, int((x1+x2)/2), y1, int((y1+y2)/2))
        v3 = lb.contains(self.domain)
        rb = Rect(int((x1+x2)/2), x2, y1, int((y1+y2)/2))
        v4 = rb.contains(self.domain)

        self.nodes = self.nodes[:idx] + [[lt,v1], [rt,v2], [lb,v3], [
    rb,v4]] + self.nodes[idx+1:]
```

Code Listing 1: The implementation of the proposed hierarchical forest-building algorithm in Python.

As shown in the training stage of overview in Section 2, the input image $x$ first goes through Gaussian smoothing and canny edge detection to get $x_e$, then the edge image $x_e$ goes through hierarchical patching to get the forest $T_x$ and an encoded sequence of length $L$, and we also use $T_x$ to encode the mask $y$ to get $y_p$. Then, we use the model to train on data with $x_p$ as input and $y_p$ as mask. After obtaining the predicted encoded sequence mask $\hat{y}_p$ for the evaluation stage, we send it to the hierarchical dispatching stage and reconstruct the prediction $\hat{y}$. Then, we can calculate the dice score between ground truth $y$ and $\hat{y}$.

We train the model using the AdamW optimizer [72] with an initial learning rate of 1e-4 over 800 epochs. The first 20 epochs are allocated for learning rate warm-up, followed by a decay by a factor of 0.1 at epochs 400 and 600. By epoch 800, the model converges. To maximize training speed, we select the largest possible batch size within the GPU memory limits. The loss function combines Dice loss and binary cross-entropy (BCE) loss:

$$
\begin{aligned}
L(\hat{y}, y) =& w \cdot L_{ce}(\hat{y}, y) + (1 - w) \cdot L_{dice}(\hat{y}, y) \\
=& - w \cdot \frac{1}{N} \sum_{i=1}^{N} [y_i \log(\hat{y}_i) + (1 - y_i) \log(1 - \hat{y}_i)] \\
& + (1 - w) \cdot (1 - \frac{2 \sum_{i=1}^{N} (\hat{y}_i \cdot y_i) + \epsilon}{\sum_{i=1}^{N} \hat{y}_i + \sum_{i=1}^{N} y_i + \epsilon})
\end{aligned}
\tag{6}
$$

where $L(\hat{y}, y)$ represents the combined loss function comprised of a weighted sum of cross-entropy (CE) loss and dice loss. The parameter $w$ controls the balance between CE loss and dice loss, set to $0.5$ in our experiments. To stabilize calculations, the smoothing term $\epsilon$ is set to $1.0$.

## B.4 Learning From the Encoded Image Space

A key concern is to ensure is that the spatial structure learned in the embeddings $\hat{y}$ corresponds to the spatial structure of the sequence $s$. We can assume that if the output prediction $\hat{y}$ can completely match the compressed mask $y_c$, then there should be a minimal loss in translation to the original mask through the spatial structure. To achieve this, we expect the regression ability of the transformer to learn the implicit spatial matching from the embedding space to the geometric pixel space, as empirically observed by a direct decrease in the loss. Since, for the positional encoding, we chose a trainable embedding vector with an initial value of zero, we speculate that if the order of patches, after hierarchical forest patching, affects the learning, then the trainable positional encoding may start to learn the positioning of tokens as manifested by the Z-order curve of the quad/octree.

## B.5 Experimental Setup

**Hardware.** All experiments were conducted on the Frontier Supercomputer [73] at ORNL. Each Frontier node is equipped with a 64-core AMD EPYC CPU and four AMD Instinct MI250X GPUs, each with 128 GB of memory. The four MI250X GPUs on a node are interconnected via AMD Infinity Fabric at 50 GB/s. Nodes are connected through the Slingshot-11 interconnect with 100 GB/s bandwidth, across a total of 9,408 nodes.

**Software.** For the software stack, we used the PyTorch 2.4 nightly build (03/16/2024), ROCm v5.7.0, MIOpen v2.19.0, and RCCL v2.13.4 with the libfabric v1.15.2 plugin. This configuration enabled efficient multi-GPU training and high-performance communication across nodes for large-scale experiments.

## B.6 Compared to Adaptive Patching Methods

A key motivation for developing SHF was to overcome the core robustness issues and significant hyperparameter sensitivity inherent in the AP approach. Rather than merely fine-tuning hyperparameters, SHF introduces a more fundamental solution by reframing hyperparameter choice as a structured, hierarchical feature extraction problem. This design choice is the key to its superior performance. We will add a table to the main paper showing the performance gains of SHF over AP, particularly at high resolutions and on the large-scale ImageNet-1K dataset. Furthermore, we will provide visualizations in the appendix that clearly demonstrate how different parameters influence the feature extraction process across the hierarchy. To demonstrate the effectiveness of SHF over AP, in high-resolution images, the following table shows the Speedup of SHF end-to-end training for the PAIP dataset at the same segmentation quality as the baseline. We use the highest dice score of the baseline models SHF and AP.

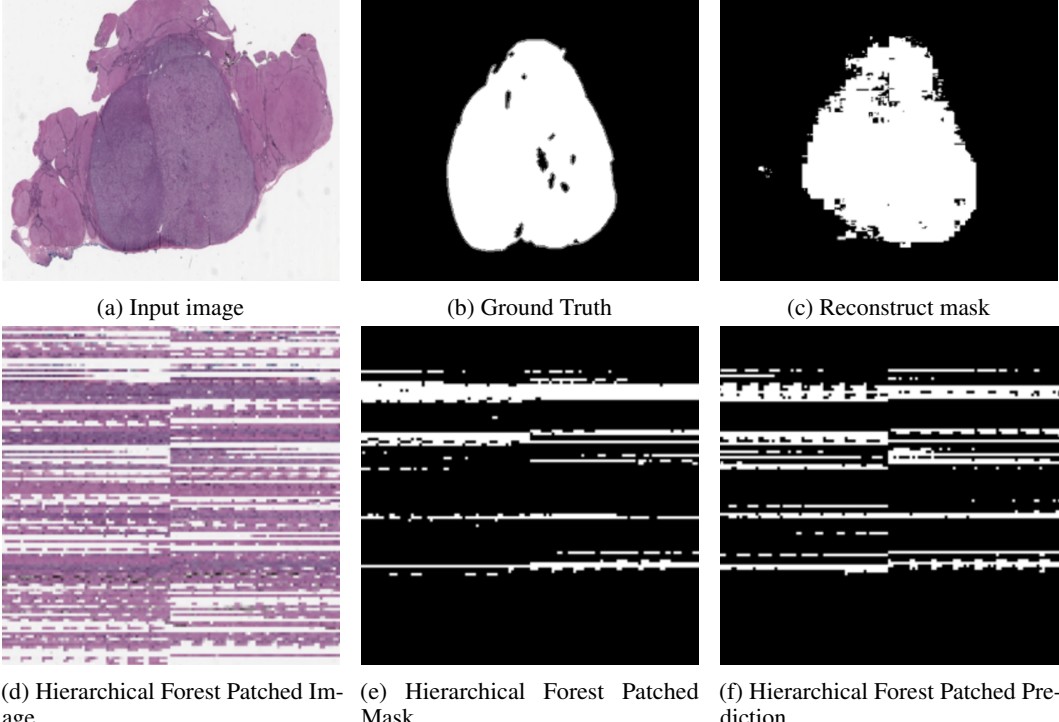

(a) Input image        (b) Ground Truth        (c) Reconstruct mask

(d) Hierarchical Forest Patched Image    (e) Hierarchical Forest Patched Mask    (f) Hierarchical Forest Patched Prediction

Figure 5: An illustrative example of an image processed using hierarchical forest patching, along with the corresponding reconstructed mask derived from the patching mask. During compression from the image domain to the sequence domain, the geometric relationships between patches are lost. Consequently, transformers must learn from sequences that do not preserve spatial structure and cannot rely on inherent geometric information.

## B.7 Generalization of SHF to Non-Medical Domains

To evaluate the generalization of SHF beyond medical imaging, we conducted experiments on the widely used ImageNet dataset, as summarized in Table 6. It is important to note that SHF was originally designed for ultra-high-resolution medical images; therefore, its performance on ImageNet is not intended to replicate state-of-the-art results. Nevertheless, we include both our results and those reported in [74] for comparison. As shown in Table 6, SHF-ViT outperforms the standard ViT by leveraging hierarchical information. However, when only a single tree with fixed hyperparameters is used, performance falls below that of the basic ViT, indicating that fixed hyperparameters can limit generalization.

Table 6: Comparison of Vision Transformer (ViT) model accuracies on ImageNet-1K.

| Model | Sequence Length | Validation Acc. | V2 Acc. |
|---|---|---|---|
| ViT-B/16 [74] | 196 | 82.2% | 72.2% |
| Tree-ViT-B/16 | 256 | 77.3% | 65.1% |
| SHF-ViT-B/16 | 196 | 82.7% | 72.8% |
| ViT-L/16 [74] | 196 | 83.0% | 72.4% |
| Tree-ViT-L/16 | 256 | 77.8% | 65.7% |
| SHF-ViT-L/16 | 196 | 83.4% | 73.1% |

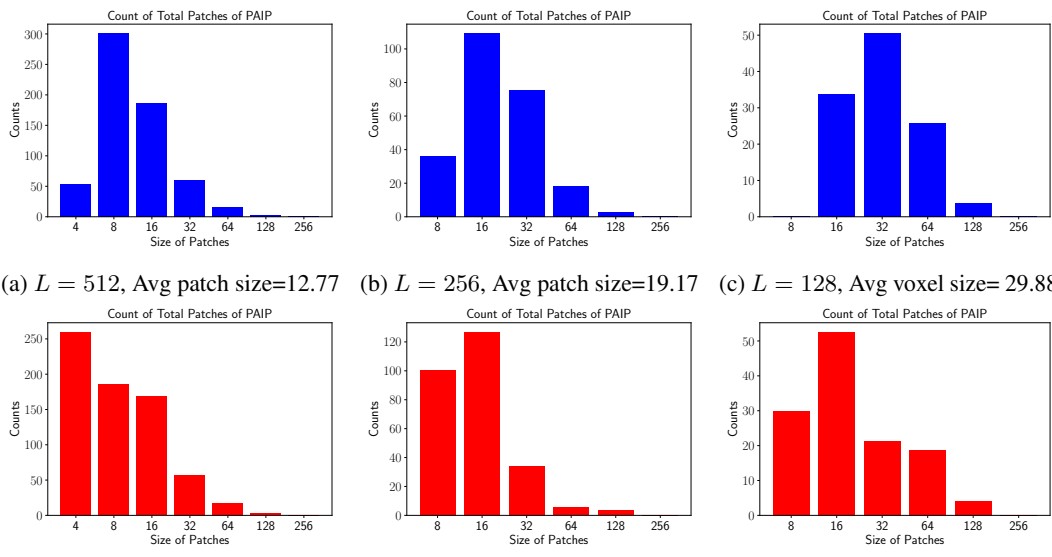

(a) $L = 512$, Avg patch size=12.77   (b) $L = 256$, Avg patch size=19.17   (c) $L = 128$, Avg voxel size= 29.88

(d) $L = 512$, Avg voxel size=6.27   (e) $L = 256$, Avg voxel size=11.79   (f) $L = 128$, Avg voxel size=17.86

Figure 6: Average patch size $[12.77, 19.17, 29.88]$ of training images with $1024$ resolution in PAIP and average octree voxel size $[6.27, 11.79, 17.86]$ of training images in KiTS lead to empirical linear scaling of the corresponding sequence length $[512, 256, 128]$. Compared to a 2D patch size in a normal distribution, a 3D voxel size distribution is preferred to small patches, which is considered a better compression ratio.

