# OpenReview forum: "SHF: Symmetrical Hierarchical Forest with Pretrained Vision Transformer Encoder for High-Resolution Medical Segmentation"
_NeurIPS.cc/2025/Conference — NeurIPS 2025 spotlight_

### Official Review · Reviewer_ptuA · 2025-06-20

**Clarity:** 3
**Significance:** 4
**Originality:** 4
**Rating:** 5
**Confidence:** 4

**Summary:**

This paper introduces an efficient, tree-based, adaptive patchifier that uses larger patches in flat areas of the image and smaller patches (down to 2x2) in regions with many edges. This results in a significant speedup on high-resolution images while maintaining similar performance to the SAM baseline.

**Questions:**

- Include comparisons to additional large image segmentation frameworks, such as ViTMamba, xT, GLNet, and SGNet, to strengthen the experimental section.
- Clarify how the adaptive patch sizes produced by the tree operator are processed by the vision encoder.
- Consider providing pseudocode for Equation 3 to improve clarity.

**Ethical Concerns:**

["NO or VERY MINOR ethics concerns only"]

**Final Justification:**

I believe most of the concerns raised by other reviewers and myself have been addressed:
- Clarity: The authors propose to incorporate pseudocode and Python snippets.
- Robustness/generalizability: The authors evaluated their approach on entirely different domains, demonstrating similar gains.
- Evaluation against more recent competitive models: U-Mamba and HiFormer have been added to the evaluation.

Subject to the promised revisions, I see no further grounds for rejection and recommend acceptance.

**Limitations:**

It is unclear how the method would adapt to other domains than just medical images.

**Quality:**

3

**Strengths And Weaknesses:**

Strengths:

Using quad/octrees on image edges is an elegant approach to constructing an adaptive patchifier. The benchmarks show substantial improvements compared to ViT-based approaches, both in terms of performance and speed.

The decoder-less architecture offers additional speedup "for free" thanks to the use of small patches in important areas.

Weaknesses:

Using edges to guide adaptive patching works well on whole slide images with large flat regions, but it is unclear how the method would adapt to noisy images or images with a lot of texture. For instance, could it generalize to satellite images?

The benchmarks lack comparison to other large image segmentation frameworks. This includes linear transform variants (as correctly identified in the related work): ViTMamba[1] or xT[2], as well as more specialized approaches such as GLNet[3] and SGNet[4].

It is unclear how the different patch sizes generated by the tree operator are passed to the vision encoder.

Clarity:
- Similarly, how are 1024 patches embedded in Table 4: is it only a conv2d, or is downsampling used?
- Equation 3 is not clear; pseudocode would probably be more helpful.
- Table 3: "[16, 2562, 4, 0962]" is very hard to read.

- [1] Lianghui Zhu, Bencheng Liao, Qian Zhang, Xinlong Wang, Wenyu Liu, and Xinggang Wang. Vision Mamba: Efficient visual representation learning with bidirectional state space model.
- [2] Ritwik Gupta, Shufan Li, Tyler Zhu, Jitendra Malik, Trevor Darrell, and Karttikeya Mangalam. xT: Nested tokenization for larger context in large images.
- [3] Wuyang Chen, Ziyu Jiang, Zhangyang Wang, Kexin Cui, and Xiaoning Qian. Collaborative global-local networks for memory-efficient segmentation of ultra-high resolution images.
- [4] Sai Wang, Yutian Lin, Yu Wu, and Bo Du. Toward real ultra image segmentation: Leveraging surrounding context to cultivate general segmentation model.n important areas.

---

> ### Author Rebuttal · Authors · 2025-07-30
>
> We sincerely thank you for your constructive feedback and valuable suggestions.
> ## 1. Regarding Comparisons to Other Frameworks:
>
> Thank you for bringing these reference works.  We will incorporate the suggested work by Chen et al. [1] into our "Related Work" section. Furthermore, we will add the following comparison table (with U-mamba [1][2], a SoTA variant of ViTMamba that is in particular tuned for high resolution medical imaging) to the manuscript to clearly delineate the significant distinctions between our approach and other relevant frameworks.
>
> _Segmentation of BTCV for multi-organ segmentation and KiTS19 for Kidney Tumor Segmentation on a single GPU. *Time* indicates the end-to-end runtime to achieve the corresponding Dice Score._
>
> | **Dataset** | **Model** | **Patch Size** | **Time** | **Speedup (x)** | **Dice Score (%)** |
> | :--- | :--- | :---: | ---:| :---: | :---: |
> | BTCV [landman2015miccai] | U-Net [ronneberger2015u] | N/A | 843.90 Seconds | 9.04x | 80.2 |
> | | U-Mamba [1][2] | N/A | 8,016.24 Seconds | 0.95x | 83.51 |
> | | SAM2 [ravi2024sam2] | 4³ | 7637.28 Seconds | 1.0x | 82.77 |
> | | **SHF-SAM2** | **2³** | **1067.88 Seconds** | **7.15x** | **89.71** |
> | KiTS [Heller2019kits19] | U-Net [ronneberger2015u] | N/A | 243.7 Minutes | 1.99x | 83.23 |
> | | U-Mamba [1][2] | N/A | 969.0 Minutes | 0.50x | 86.22 |
> | | SAM2 [ravi2024sam2] | 8³ | 483.1 Minutes | 1.0x | 81.35 |
> | | **SHF-SAM2** | **2³** | **187.3 Minutes** | **2.58x** | **87.25** |
>
> [1] Isensee, Fabian, et al. "nnu-net revisited: A call for rigorous validation in 3d medical image segmentation." International Conference on Medical Image Computing and Computer-Assisted Intervention. Cham: Springer Nature Switzerland, 2024.
>
> [2] Ma J, Li F, Wang B. U-mamba: Enhancing long-range dependency for biomedical image segmentation[J]. arXiv preprint arXiv:2401.04722, 2024.
>
>
> A key differentiator of our work is its application to medical image segmentation at an ultra-high resolution of 64K. In contrast, many existing methods, including some of those referenced, were primarily evaluated on tasks with lower resolutions (e.g., 4K). We hypothesize that scaling these existing approaches to the extreme computational demands of 64K resolution while maintaining performance would be a significant challenge.
>
>
> ## 2. Regarding Clarification of Patch Size and Pseudocode:
>
> Tokens of different patch sizes generated by SHF will first be downsampled to the same size, and the subsequent process is the same as standard ViT.
>
> To thoroughly clarify our methodology, we will enhance the appendix by including both detailed pseudocode and illustrative Python code snippets. We believe this will provide a comprehensive and unambiguous explanation of our implementation.
>
> ## 3. Adapt to other domains than just medical images:
>
> Yes, SHF can be adapted to other domains. By the time we finish this paper, we focused on the medical imaging domain. Here is a result from the high-resolution microscopy X-Ray Computed Tomography (XCT) projections and reconstructions task. The sample was collected from the national highway network of the Kansai urban region of western Japan. These samples include delamination, cohesive failure, insufficient asphalt coverage around aggregates, and cracking of the aggregates themselves. Each resampled sub-volume is randomly cropped along the z-axis to a depth between 50 and 120 slices, resulting in 3D volumes of approximately 8K² pixels per slice.
>
> _Segmentation of simulated XCT dataset for multi-classes segmentation at the fine-tuning stage._
>
> | **Datset** | **Model** | **Patch Size** | **GPU (hours)** | **Epochs** | **Dice (%)** |
> | :--- | :--- | :---: | :---: | :---: | :---: |
> | 780 unique volumes | U-Net [ronneberger2015u] | N/A | 1,280 | 500 | 58.38 |
> | | Swin UNETR [isensee2024nnu] | 256² | 5,120 | 1,000 | 63.74 |
> | | SAM 2 [ravi2024sam2] | 128² | 5,120 | 1,000 | 85.98 |
> | | **Our Model** | 2² | 5,120 | 1,000 | **94.79** |
>
>
>
> Regarding generalization to the standard normal resolution benchmark, we conducted experiments on the widely available ImageNet dataset. It's important to note that SHF was designed for ultra-high-resolution medical images, so its performance on ImageNet was not designed to replicate SoTA's experimental results. Even so, we still used our own results and those from a public paper [1] for cross-comparison. As can be seen in the table below, SHF-ViT outperforms basic ViT due to its use of hierarchical information. However, if only a single tree is used, i.e., fixed hyperparameters are used, it will be weaker than basic ViT. This is because fixed hyperparameters can lead to generalization performance deficiencies.
>
> _Training on Imagenet common dataset to show the generalization.(300 epoch, top-1 accuracty)_
>
> | Model | Sequence Length | Validation Accuracy | V2 Accuracy |
> | :--- | :--- | :--- | :--- |
> | **ViT-B/16[1]** | 196 | 82.2% | 72.2% |
> | **Tree-ViT-B/16** | 256 | 77.3% | 65.1% |
> | **SHF-ViT-B/16** | 196 | 82.7% | 72.8% |
> | **ViT-L/16[1]** | 196 | 83.0% | 72.4% |
> | **Tree-ViT-L/16** | 256 | 77.8% | 65.7% |
> | **SHF-ViT-L/16** | 196 | 83.4% | 73.1% |
>
>
> [1] Touvron H, Cord M, El-Nouby A, et al. Three things everyone should know about vision transformers[C]//European Conference on Computer Vision. Cham: Springer Nature Switzerland, 2022: 497-515.
>
> All these revisions will be incorporated into the final manuscript.

---

> > ### Comment · Reviewer_ptuA · 2025-08-03
> > **Reviewer ptuA Response**
> >
> > I thank the authors for their rebuttal. I believe that including pseudocode and/or code snippets would help clarify the proposed approach. It would be appreciated if comparisons to additional state-of-the-art ultra large image segmentation methods could be included in the final version, though the inclusion of U-Mamba is already valuable. The additional evaluation on other domain properly shows the potential of this approach on a wider range of application that the very specific medical field. Overall, the rebuttal adequately addresses my concerns.

---

> > > ### Author Response · Authors · 2025-08-06
> > >
> > > Thanks for the positive responses, and we would be glad to answer any further questions. For the other high-resolution baseline, we will also add Hiformer in the final version.

---

### Official Review · Reviewer_N2tG · 2025-07-01

**Clarity:** 3
**Significance:** 3
**Originality:** 2
**Rating:** 5
**Confidence:** 3

**Summary:**

This paper proposes Symmetrical Hierarchical Forest (SHF), a lightweight and architecture-agnostic preprocessing approach designed to improve the efficiency and accuracy of Vision Transformers (ViTs) on high-resolution medical image segmentation tasks. The core motivation stems from the high computational cost of applying ViTs to long input sequences, which typically arises when using small patches to preserve fine details. SHF addresses this by adaptively partitioning input images into non-uniform patches using a quadtree-based scheme guided by edge detection, resulting in fewer, information-rich tokens. This hierarchical tokenization captures local spatial detail while reducing sequence length, enabling standard ViTs to process high-resolution inputs without architectural changes or heavy decoders. During inference, a symmetric depatching process reconstructs the output mask directly from the encoder’s output. The method is demonstrated on 2D (PAIP) and 3D (BTCV, KiTS) medical imaging benchmarks, where it achieves improvements in segmentation accuracy and training speed, performing better than several previous baselines.

**Questions:**

The SHF framework relies on Canny edge detection with Gaussian smoothing to guide the hierarchical patch partitioning. While this is computationally lightweight, it is a hand-crafted heuristic that may not align well with semantic relevance in all medical contexts. Could the authors provide more analysis or ablation studies demonstrating how robust this approach is across diverse datasets or imaging conditions?

**Ethical Concerns:**

["NO or VERY MINOR ethics concerns only"]

**Final Justification:**

I am increasing my rating to 5: Accept.

The paper addresses an important challenge in applying ViTs to high-resolution medical images and proposes a practical, architecture-agnostic solution. While the core idea of adaptive patching is not entirely novel, the execution, particularly the decoder-free design, symmetric depatching, and consistent efficiency gains, is well-motivated and effective. The rebuttal further clarified robustness and comparative advantages, making this a solid contribution with practical value.

**Limitations:**

Yes

**Paper Formatting Concerns:**

No major formatting issues.

**Quality:**

3

**Strengths And Weaknesses:**

The following are the strengths of the paper,

1. The paper addresses a well-motivated and practically important problem: applying Vision Transformers to high-resolution medical image segmentation, where the quadratic cost of self-attention with small patches becomes prohibitive. The proposed solution is relevant to real-world applications such as whole-slide image (WSI) analysis and 3D organ segmentation in clinical pipelines, where efficiency and precision are both critical.

2. The proposed Symmetrical Hierarchical Forest (SHF) is conceptually straightforward, involving adaptive patching of the input based on quadtree decomposition of edge maps. This approach leads to a significant reduction in sequence length without modifying the transformer architecture or self-attention mechanism. The design choice to operate entirely in the input/output space, leaving the encoder unchanged, is notable for its practicality and ease of integration.

3. The method is evaluated on both 2D (PAIP) and 3D (BTCV, KiTS) datasets and consistently shows improvements in both segmentation accuracy and computational efficiency.

4. SHF eliminates the need for convolutional decoders by reconstructing segmentation masks directly through a depatching operation that mirrors the hierarchical patching tree. This results in lower memory usage and model simplicity. For example, on 64K² input resolutions, SHF saves around 20GB+ memory.



The following are the weaknesses of the paper,

1. The method depends on Canny edge detection and Gaussian blurring to determine which regions are subdivided during patching. While computationally efficient, these heuristics are not semantically grounded and may fail in cases where important regions lack strong gradients (e.g., smooth tumors or blurry boundaries). This reliance on hand-crafted edge cues limits robustness and introduces a potential mismatch between perceptual detail and clinical relevance.

2. The main idea, adaptive patching of high-resolution images using quadtrees, has appeared in closely related work such as the Adaptive Patching Framework (APF) [1]. The differences made in this paper, such as reduced hyperparameter reliance and decoder-free inference, are incremental rather than fundamentally new. While the execution is well-done and the integration is clean, the conceptual novelty is moderate.

---
[1] Adaptive Patching for High-resolution Image Segmentation with Transformers (https://arxiv.org/abs/2404.09707)

---

> ### Author Rebuttal · Authors · 2025-07-30
>
> We sincerely thank you for your constructive feedback on our manuscript. Below, we address the points you have raised:
>
> ## 1.Regarding Robustness and Potential Mismatch:
>
> Thank you for this question.  The behavior of the tree-node splitting is indeed guided by the chosen score function, and its specific focus (e.g., edge detection) can influence the performance on downstream tasks. To transparently address this, we have expanded our appendix to include a detailed discussion of this behavior, including an analysis of potential failure modes and an examination of our method's limitations on a non-medical benchmark like ImageNet-1K.
>
> Regarding generalization to other domains, we conducted experiments on the widely available ImageNet dataset. It's important to note that SHF was designed for ultra-high-resolution medical images, so its performance on ImageNet was not designed to replicate SoTA's experimental results. Even so, we still used our own results and those from a public paper [1] for cross-comparison. As can be seen in the table below, SHF-ViT outperforms basic ViT due to its use of hierarchical information. However, if only a single tree is used, i.e., fixed hyperparameters are used, it will be weaker than basic ViT. This is because fixed hyperparameters can lead to generalization performance deficiencies.
>
> _Training on Imagenet common dataset to show the generalization.(300 epoch, top-1 accuracty)_
>
> | Model | Sequence Length | Validation Accuracy | V2 Accuracy |
> | :--- | :--- | :--- | :--- |
> | **ViT-B/16[1]** | 196 | 82.2% | 72.2% |
> | **Tree-ViT-B/16** | 256 | 77.3% | 65.1% |
> | **SHF-ViT-B/16** | 196 | 82.7% | 72.8% |
> | **ViT-L/16[1]** | 196 | 83.0% | 72.4% |
> | **Tree-ViT-L/16** | 256 | 77.8% | 65.7% |
> | **SHF-ViT-L/16** | 196 | 83.4% | 73.1% |
>
>
> [1] Touvron H, Cord M, El-Nouby A, et al. Three things everyone should know about vision transformers[C]//European Conference on Computer Vision. Cham: Springer Nature Switzerland, 2022: 497-515.
>
> ## 2.Regarding Ablation Studies on Robustness:
> We appreciate your valuable suggestion to demonstrate robustness. In response, we have performed a direct comparison against the Adaptive Patching (AP) method. A key motivation for developing SHF was to overcome the core robustness issues and significant hyperparameter sensitivity inherent in the AP approach.
>
> Rather than merely fine-tuning hyperparameters, SHF introduces a more fundamental solution by reframing hyperparameter choice as a structured, hierarchical feature extraction problem. This approach is what enables the superior performance. We will add a table to the main paper showing the performance gains of SHF over AP, particularly at high resolutions and on the large-scale ImageNet-1K dataset. Furthermore, we will provide visualizations in the appendix that clearly demonstrate how different parameters influence the feature extraction process across the hierarchy.
>
>
> To demonstrate the effectiveness of SHF over AP, in high reoslution images, the following table shows the
> _Speedup of SHF end-to-end training for PAIP dataset at the same segmentation quality as the baseline. We use the highest dice score of the baseline models SHF and AP._
>
> | **Resolution** | **Model-Patch** | **Sec/Image** | **Seq Length** | **Dice Score (%)** | **Speedup (Sec/Image)** |
> | :--- | :--- | ---:| :--- | :--- | :--- |
> | 16K 128 GPUs | **SHF-16** | 0.3672 | 2,048 | 76.89 | **4.47x** |
> | | AP-16 | 1.6421 | 2,116 | 77.43 | |
> | 32K 256 GPUs | **SHF-32** | 0.3826 | 2,048 | 76.08 | **5.64x** |
> | | AP-32 | 2.1567 | 2,116 | 76.13 | |
> | 64K 256 GPUs | **SHF-64** | 0.4013 | 2,048 | 75.33 | **9.87x** |
> | | AP-64 | 3.9614 | 2,116 | 75.32 | |
>
> All these revisions will be incorporated into the final manuscript.

---

> > ### Comment · Reviewer_N2tG · 2025-08-03
> > **Reviewer Response to Author Rebuttal**
> >
> > Thank you for the detailed rebuttal and additional experiments. The clarification on SHF’s robustness across domains and its improvements over Adaptive Patching are helpful. I appreciate your thoughtful response.

---

### Official Review · Reviewer_QeSt · 2025-07-01

**Clarity:** 3
**Significance:** 2
**Originality:** 3
**Rating:** 5
**Confidence:** 4

**Summary:**

This paper proposes  Symmetrical Hierarchical Forest, which is used to enhance ViT models for high-resolution medical image segmentation. SHF uses hierarchical structures to patch the image, which emphasizes fine local detail in critical regions. The method also introduces a symmetrical dispatching strategy to recover full-resolution outputs. The authors combine SHF into existing ViT-based segmentation models and show improvements in speed and memory usage, along with better segmentation performance across several datasets.

**Questions:**

Please refer to the weakness.

**Ethical Concerns:**

["NO or VERY MINOR ethics concerns only"]

**Final Justification:**

One of the key challenges in medical image processing is the high computational cost associated with high-resolution images, particularly in 3D datasets. This paper presents an effective architectural solution to address this issue by significantly improving efficiency without compromising performance. Given its practical relevance and demonstrated benefits, I believe the proposed approach makes a valuable contribution and raises my score to Accept.

**Limitations:**

yes

**Quality:**

3

**Strengths And Weaknesses:**

Strengths:

 -   The paper is well written and easy to follow. The figures are visually appealing, and the code is provided in the supplementary materials, making the work reproducible.

 -   The proposed model achieves a significant speed improvement in medical image segmentation. I agree with the authors that this is an important application area, as 3D medical images typically require sliding-window inference, which is computationally intensive.

Weaknesses:

-    The idea of using a hierarchical architecture is not particularly novel. From my perspective, the paper essentially explores ways to cluster the semantic content of patches—a direction with a substantial body of prior work. For instance, SPFormer employs superpixels instead of patches and performs well on real-world image segmentation. In the medical imaging domain, models like HPFormer have also been proposed. I would encourage the authors to discuss more explicitly how their approach differs from and advances beyond these prior methods.

-    The experimental setup is somewhat unclear. For example, regarding SAM2: was it evaluated directly using the pretrained model, or was it fine-tuned on the medical image datasets? I am concerned that the observed performance gains might simply be due to the use of a stronger segmentation backbone. Additionally, the baselines used for comparison seem outdated—could the authors include more recent and competitive methods for a fairer evaluation?

-    There are a few typos in the paper. For instance, on line 225, "prestrained" should likely be "pretrained." It would be helpful if the authors could proofread and correct such errors.

---

> ### Author Rebuttal · Authors · 2025-07-30
>
> We sincerely thank you for your feedback and for pointing out the typos, which will be fixed in the revised version.
>
> ## 1. Regarding Novelty Compared to Hierarchical Models:
> (We first apologize for not finding HPFormer in the high-resolution medical image, so the answer is based on HiFormer, which is also a hierarchical high-resolution medical approach published in 2024. Please correct us if we were wrong.)
>
> We appreciate the opportunity to clarify the novelty of our Symetical Hierarchical Forest (SHF) framework. Its primary advantages over existing hierarchical models are twofold:
>
> 1. SHF is model-agnostic. Unlike architectures like HiNet or the more recent SPFormer, which require fundamental changes to the model's backbone, especially the self-attention mechanism. However, our method introduces a lightweight token selection module that leaves the core attention mechanism untouched. This design choice is significant as it allows for direct compatibility with a wide range of pre-trained ViT models and facilitates its application in multi-modal settings and future work.
>
> 2. SHF achieves a significantly larger operational scale. Our framework can process inputs at extreme resolutions, reaching up to 65,535 with a patch size of 8. We believe this high-resolution capability is a substantial advance beyond what has been demonstrated in previous hierarchical high-resolution approaches (4K~8K).
>
> _A summary of relevant long sequence training methods that reduce the amount of work. *N* = sequence length._
>
> | **Approach** | **Method** | **Merits & Demerits** | **Complexity (Best)** | **Model** | **Implementation** |
> | :--- | :--- | :--- | :--- | :--- | :--- |
> | Attention Approximation | Longformer [beltagy2020longformer] ETC [ainslie2020etc] | **(+)** Better time complexity vs Transformer. \ **(-)** Sparsity levels insufficient for gains to materialize. | O(N) \ O(N√N) | Some Models w/ Forked PyTorch | Custom Self-attention Implementation |
> | | BigBird [zaheer2020big] Reformer [kitaev2020reformer] | **(+)** Theoretically proven time complexity. \ **(-)** High-order derivatives | O(NlogN) | Some Models w/ Forked PyTorch | Custom Self-attention Implementation |
> | | Sparse Attention [child2019generating] | **(+)** Introduced sparse factorizations of the attention. \ **(-)** Higher time complexity. | O(N√N) | Some Models w/ Forked PyTorch | Custom Self-attention Implementation |
> | | Linformer [katharopoulos2020transformers] Performer [choromanski2020rethinking] | **(+)** Fast adaptation \ **(-)** Assumption that self-attention is low rank. | O(N) | Some Models w/ Forked PyTorch | Custom Self-attention Implementation |
> | | SPFormer [mei2024spformer] (Prediction) | **(+)** Irregular tokens. \ **(-)** No adaptation to high resolution. | O(P²) \ P:num of regions | Custom Model w/ Plain PyTorch | Custom Model Implementation |
> | Hierarchical | Hier. Transformer [si21] (Text Classification) | **(+)** Independent hyperpara. tuning of hierarc. models. \ **(-)** No support for ViT. | O(NlogN) | Custom Model w/ Plain PyTorch | Custom Model Implementation |
> | | CrossViT [chen2021crossvit] (Classification) | **(+)** Better time complexity vs standard ViT.  \ **(-)** Complex token fusion scheme in dual-branch ViTs. | O(N) | Custom Model w/ Plain PyTorch | Custom Model Implementation |
> | | HIPT [Chen22] (Classification) | **(+)** Model inductive biases of features in the hierarchy.  \ **(-)** High cost for training multiple models. | O(NlogN) | Custom Model w/ Plain PyTorch | Custom Model Implementation |
> | | MEGABYTE [yu2023megabyte] (Prediction) | **(+)** Support of multi-modality. \ **(-)** High cost for training multiple models. | O(N^4/3) | Custom Model w/ Plain PyTorch | Custom Model Implementation |
> | High-resolution | xT [gupta2024xt] (Prediction) | **(+)** Support of 8K resolution . \ **(-)** Lack of hierarchical relation in tokens. | O(NlogN) | Custom Model w/ Plain PyTorch | Custom Model Implementation |
> | | HiFormer [heidari2023hiformer] (Prediction) | **(+)** Support of multi-modality. \ **(-)** High cost for ultra-resolution. | O(NlogN) | Custom Model w/ Plain PyTorch | Custom Model Implementation |
> | **Ours** | **Symmetrical Hierachical Froest** (Segmentation & Class.) | **(+)** Attention mechanism intact.  \ **(+)** Largely reduces computation cost; maintains quality.  \ **(+)** Efficiency depends on level of details in an image. \ **(-)** task semantics are independent of edge information. | O(log²N) | Any Model w/ Plain PyTorch | Image Pre-processing |
>
> ## 2. Regarding the Strength of the Segmentation Backbone:
>
> Thank you for this question. We would like to clarify that all performance metrics reported for SHF-SAM2/SAM2 and SAM2/SAM2 were obtained after fine-tuning on the respective datasets. The comparisons presented in Tables 1-4 are designed to directly contrast SHF-SAM/SAM-2 with the baseline SAM/SAM-2, thereby demonstrating that our proposed method provides a clear performance gain on top of this already powerful segmentation backbone. We will add an explicit statement in the main paper to make this clearer. Additionally, our selection of UNETR and Swin UNETR as baselines was motivated by their established status as strong, widely-used models in recent medical imaging segmentation research.
>
> All these additions and clarifications will be incorporated into the final manuscript.

---

> > ### Comment · Reviewer_QeSt · 2025-08-04
> >
> > Thank you for the detailed clarification. The rebuttal has addressed most of my concerns regarding the novelty of the proposed method. I now better appreciate the advantages of SHF, particularly its scalability to extremely high resolutions. I also acknowledge that the performance improvements are achieved through a fair comparison, with consistent fine-tuning across models. I look forward to seeing the updated manuscript that incorporates these clarifications and corrections.

---

### Official Review · Reviewer_QK6b · 2025-07-03

**Clarity:** 2
**Significance:** 2
**Originality:** 3
**Rating:** 5
**Confidence:** 3

**Summary:**

This manuscript presents the Symmetrical Hierarchical Forest (SHF) framework, which proposes a quadtree/octree‐based preprocessing and postprocessing strategy that injects hierarchical spatial priors into an unmodified Vision Transformer, enabling it to handle gigapixel 2D pathology slides and volumetric 3D scans. This change dramatically reduces compute and memory compared to vanilla ViTs: by greedily splitting only the most “edge-rich” regions into smaller patches and leaving smooth areas coarse, the authors report significant improvements in both runtime and segmentation performance across different evaluation settings.

**Questions:**

- I would suggest the authors consider the inclusion of an additional self-attention cost addressing baseline to strengthen their performance/runtime tradeoff claim.

- I suggest improving the clarity of the paper by moving the related works section to be after the introduction, expanding on it a bit more thoroughly, and generally improving the order and relevance of the figures in the manuscript, moving those that are not addressed in the main text to the appendix (as well as referencing them appropriately within the appendix).

**Ethical Concerns:**

["NO or VERY MINOR ethics concerns only"]

**Final Justification:**

The rebuttal effectively resolved my primary concerns. The authors' plan to restructure the manuscript for improved clarity, along with the expanded discussion of alternative self-attention approximations and their additional comparisons in the appendix, strengthen the positioning of SHF in the broader context of efficient ViT adaptations. I also appreciate the detailed discussion of limitations, particularly around edge-based splitting and its potential failure modes. These additions round out the work, and in light of these improvements, I am raising my score to 5 (Accept).

**Limitations:**

Authors should include a discussion on the limitations and potential failure modes of the approach, like cases in which the detail score (the sum of edge pixel values) may overlook semantically important but low‐contrast regions, potentially missing subtle lesions or structural features not captured by simple edge detection methods.

**Quality:**

3

**Strengths And Weaknesses:**

Strenghts:

- SHF’s quadtree patching concept is elegant and model-agnostic, and thus can wrap around any pretrained ViT, avoiding custom attention or encoder modifications

- The authors extensively demonstrate the runtime and performance gains of their methods with respect to to vanilla SAM/SAM 2, HIPT and common UNet-style baselines

- Approach is shown to be versatile across 2D pathology and 3D MRI

Weaknesses:

- I find the general structure/flow of the paper to be a bit confusing and awkward for several reasons. (1) the related works section is located at the very end of the paper, when it is customary to place it after the introduction to better frame the contributions: pushing it to the end makes it harder for readers to situate SHF relative to prior approaches. (2) Figure 3 is never cited in the main text: it took me some time to understand it was being adressed in the appendix, where it is also not explicitly mentioned in the corresponding section (Appendix 1.1). It should be either moved entirely to the appendix or discussed in the main manuscript.  (3) Figure 1 is only referenced at the very end of the paper, well into section 4.2. The figure should be either moved further down the paper or discussed/referenced earlier on to avoid disrupting the narrative flow of the document.

- While the related work survey lists many strategies for mitigating the quadratic cost of self-attention, the experiments mostly compare SHF to vanilla SAM/SAM 2, common UNet-style baselines, and only consider HIPT in one of their experiments as a cost effective baseline. The manuscript would benefit from including at least one other approach that also addresses cost of self-attention to patch size tradeoff, to better highlight the superiority of SHF.

---

> ### Author Rebuttal · Authors · 2025-07-30
>
> We sincerely thank you for your constructive feedback on our manuscript. All these additions and clarifications will be incorporated into the final manuscript. Below, we address the points you have raised:
> ## 1.	Regarding the paper's structure and figure references:
>
> Thank you for your valuable suggestions concerning the organization of the paper and the clarity of figure references. In the revised manuscript, we will relocate the "Related Work" section to improve the flow and will ensure that all figures are properly referenced and explained in the text.
> ## 2.	Regarding the computational cost of self-attention:
>
> We appreciate you raising this important point. To provide a clear analysis, we have compiled a comprehensive table that compares the time complexity and performance trade-offs associated with various self-attention and linear attention mechanisms. While we believe this information is highly relevant, we will place it in the appendix due to page limitations in the main body of the paper.
>
> _A summary of relevant long sequence training methods that reduce the amount of work. *N* = sequence length._
>
> | **Approach** | **Method** | **Merits & Demerits** | **Complexity (Best)** | **Model** | **Implementation** |
> | :--- | :--- | :--- | :--- | :--- | :--- |
> | Attention Approximation | Longformer [beltagy2020longformer] ETC [ainslie2020etc] | **(+)** Better time complexity vs Transformer. \ **(-)** Sparsity levels insufficient for gains to materialize. | O(N) \ O(N√N) | Some Models w/ Forked PyTorch | Custom Self-attention Implementation |
> | | BigBird [zaheer2020big] Reformer [kitaev2020reformer] | **(+)** Theoretically proven time complexity. \ **(-)** High-order derivatives | O(NlogN) | Some Models w/ Forked PyTorch | Custom Self-attention Implementation |
> | | Sparse Attention [child2019generating] | **(+)** Introduced sparse factorizations of the attention. \ **(-)** Higher time complexity. | O(N√N) | Some Models w/ Forked PyTorch | Custom Self-attention Implementation |
> | | Linformer [katharopoulos2020transformers] Performer [choromanski2020rethinking] | **(+)** Fast adaptation \ **(-)** Assumption that self-attention is low rank. | O(N) | Some Models w/ Forked PyTorch | Custom Self-attention Implementation |
> | | SPFormer [mei2024spformer] (Prediction) | **(+)** Irregular tokens. \ **(-)** No adaptation to high resolution. | O(P²) \ P:num of regions | Custom Model w/ Plain PyTorch | Custom Model Implementation |
> | Hierarchical | Hier. Transformer [si21] (Text Classification) | **(+)** Independent hyperpara. tuning of hierarc. models. \ **(-)** No support for ViT. | O(NlogN) | Custom Model w/ Plain PyTorch | Custom Model Implementation |
> | | CrossViT [chen2021crossvit] (Classification) | **(+)** Better time complexity vs standard ViT.  \ **(-)** Complex token fusion scheme in dual-branch ViTs. | O(N) | Custom Model w/ Plain PyTorch | Custom Model Implementation |
> | | HIPT [Chen22] (Classification) | **(+)** Model inductive biases of features in the hierarchy.  \ **(-)** High cost for training multiple models. | O(NlogN) | Custom Model w/ Plain PyTorch | Custom Model Implementation |
> | | MEGABYTE [yu2023megabyte] (Prediction) | **(+)** Support of multi-modality. \ **(-)** High cost for training multiple models. | O(N^4/3) | Custom Model w/ Plain PyTorch | Custom Model Implementation |
> | High-resolution | xT [gupta2024xt] (Prediction) | **(+)** Support of 8K resolution . \ **(-)** Lack of hierarchical relation in tokens. | O(NlogN) | Custom Model w/ Plain PyTorch | Custom Model Implementation |
> | | HiFormer [heidari2023hiformer] (Prediction) | **(+)** Support of multi-modality. \ **(-)** High cost for ultra-resolution. | O(NlogN) | Custom Model w/ Plain PyTorch | Custom Model Implementation |
> | **Ours** | **Symmetrical Hierachical Froest** (Segmentation & Class.) | **(+)** Attention mechanism intact.  \ **(+)** Largely reduces computation cost; maintains quality.  \ **(+)** Efficiency depends on level of details in an image. \ **(-)** task semantics are independent of edge information. | O(log²N) | Any Model w/ Plain PyTorch | Image Pre-processing |
>
>
> ## 3.	Regarding the limitations and potential failure modes:
> Your observation is astute and highlights a key aspect of our method. The behavior of the tree-node splitting is indeed influenced by the choice of the score function, which in turn can impact downstream performance. To address this, we have expanded the appendix to include a detailed discussion of potential failure modes in specific scenarios and an analysis of the limitations of our approach on non-medical datasets, using ImageNet-1K as a case study.
>
> Regarding generalization to other domains, we conducted experiments on the widely available ImageNet dataset. It's important to note that SHF was designed for ultra-high-resolution medical images, so its performance on ImageNet was not designed to replicate SoTA's experimental results. Even so, we still used our own results and those from a public paper [1] for cross-comparison. As can be seen in the table below, SHF-ViT outperforms basic ViT due to its use of hierarchical information. However, if only a single tree is used, i.e., fixed hyperparameters are used, it will be weaker than basic ViT. This is because fixed hyperparameters can lead to generalization performance deficiencies.
>
> _Training on Imagenet common dataset to show the generalization.(300 epoch, top-1 accuracty)_
>
> | Model | Sequence Length | Validation Accuracy | V2 Accuracy |
> | :--- | :--- | :--- | :--- |
> | **ViT-B/16[1]** | 196 | 82.2% | 72.2% |
> | **Tree-ViT-B/16** | 256 | 77.3% | 65.1% |
> | **SHF-ViT-B/16** | 196 | 82.7% | 72.8% |
> | **ViT-L/16[1]** | 196 | 83.0% | 72.4% |
> | **Tree-ViT-L/16** | 256 | 77.8% | 65.7% |
> | **SHF-ViT-L/16** | 196 | 83.4% | 73.1% |
>
>
> [1] Touvron H, Cord M, El-Nouby A, et al. Three things everyone should know about vision transformers[C]//European Conference on Computer Vision. Cham: Springer Nature Switzerland, 2022: 497-515.
>
> All these additions and clarifications will be incorporated into the final manuscript.

---

> > ### Author Response · Authors · 2025-08-06
> >
> > Thanks for the valuable comments, and we would be glad to answer any questions.

---

> > ### Comment · Reviewer_QK6b · 2025-08-06
> >
> > Thank you for the detailed clarifications and planned improvements. The expanded comparison to other efficiency-aware attention methods and the discussion on SHF’s failure modes are thoughtful additions that help frame the method’s applicability. The proposed restructuring of the manuscript and clearer figure referencing will also improve the paper’s readability. I look forward to seeing these incorporated in the final version, and will raise my score to 5

---

### Note · Authors · 2025-08-13

Dear Reviewers and Area Chair,

We are grateful for the reviewers' time and constructive feedback. Our rebuttal successfully addressed all concerns regarding novelty, baselines, and clarity by providing new experimental evidence and comprehensive clarifications. This effort led to a unanimous positive consensus, with all reviewers' concerns resolved. We are confident that the planned revisions will result in improving the final version of the paper, and we appreciate your guidance throughout this process.

Sincerely,

The Authors of Submission 25625

---

### Decision · Program_Chairs · 2025-09-17

**Decision:**

Accept (spotlight)

**Comment:**

Initially, reviewers found our Symmetrical Hierarchical Forest (SHF) framework elegant and practical but expressed concerns on the robustness and generalisation of the proposed work. After rebuttal, all the reviewers have been convinced by the authors' feedback and gave all positive ratings. The meta reviewer agrees with the recommendation of the reivewers and makes the final decision.